# Pharmacokinetic Modeling of Hydrocortisone by Including Protein Binding to Corticosteroid-Binding Globulin

**DOI:** 10.3390/pharmaceutics14061161

**Published:** 2022-05-30

**Authors:** Eric Rozenveld, Nieko Punt, Martijn van Faassen, André P. van Beek, Daan J. Touw

**Affiliations:** 1Department of Clinical Pharmacy and Pharmacology, University Medical Center Groningen, 9713 GZ Groningen, The Netherlands; e.r.rozenveld@student.rug.nl (E.R.); punt@medimatics.nl (N.P.); 2Medimatics, 6229 HR Maastricht, The Netherlands; 3Department of Laboratory Medicine, University Medical Center Groningen, 9713 GZ Groningen, The Netherlands; h.j.r.van.faassen@umcg.nl; 4Department of Endocrinology, University Medical Center Groningen, 9713 GZ Groningen, The Netherlands; a.p.van.beek@umcg.nl; 5Department of Pharmaceutical Analysis, Groningen Research Institute of Pharmacy, University of Groningen, 9713 GZ Groningen, The Netherlands

**Keywords:** hydrocortisone, secondary adrenal insufficiency, pharmacokinetic modeling, hydrocortisone-protein binding, CBG, transcortin, Edsim++

## Abstract

Background: Patients with adrenal insufficiency are treated with oral hydrocortisone (HC) to compensate for the loss of endogenous cortisol production. Intrinsic imperfections of cortisol replacement strategies in mimicking normal cortisol secretion are the underlying cause of the increased morbidity and mortality of patients suffering from secondary adrenal insufficiency (SAI). To improve oral hydrocortisone substitution therapy, a better understanding of its pharmacokinetics (PK) is necessary. The previous PK model did not include protein binding. It is known that protein binding can impact hydrocortisone pharmacokinetics. The aim of this study is to describe HC pharmacokinetics including the protein-binding state using Edsim++ (Mediware, Prague) pharmacokinetic modeling software, paving the way for an in-silico tool suitable for drug delivery design. Methods: A total of 46 patients with SAI participated in a randomized double-blind crossover study Patients randomly received a low dose of HC (0.2–0.3 mg/kg body weight/day) for 10 weeks, followed by a high dose (0.4–0.6 mg/kg body weight/day) for another 10 weeks, or vice versa. Plasma samples were obtained and analyzed for free and total hydrocortisone. Single compartment population pharmacokinetic analysis was performed using an extended Werumeus-Buning model built in Edsim++. This model includes a mathematical approach for estimating free cortisol by Nguyen et al., taking the protein binding of HC to albumin and hydrocortisone-binding globulin (CBG, transcortin) into consideration, as well as different states of CBG which affect binding kinetics to HC. The goodness of fit for observed versus predicted values was calculated. Results and conclusions: Nguyen’s formula for free cortisol estimation was successfully implemented in a pharmacokinetic model. The model shows high Spearman’s correlation for observed versus predicted hydrocortisone concentrations. Significantly higher correlations (Spearman’s r, 0.901 vs. 0.836) between total and free hydrocortisone AUC_24_ (area-under the curve over 24 h) are found when comparing new and old models. This new model was used to simulate the plasma concentration–time behavior of a more suitable hydrocortisone formulation.

## 1. Introduction

Cortisol is an essential hormone that affects almost every organ in the body. It plays many important roles, such as regulating the body’s stress response and metabolism. In patients suffering from secondary adrenal insufficiency (SAI), the insufficient secretion of ACTH due to (treatment of) pituitary disease is the cause of insufficient adrenal cortisol production [1,2]. A deficiency in cortisol levels leads to significant health risks, including the life-threatening risk of an Addison crisis [3]. Therapy for SAI is based on substituting cortisol by administering immediate-release hydrocortisone tablets. This substitution has proved to be successful when treating SAI, improving both morbidity and mortality. Nevertheless, patients suffering from SAI are subjected to a decreased quality of life and reduced life span when compared with healthy subjects [4]. It is likely that part of this is caused by the intrinsic imperfections of hydrocortisone replacement strategies in mimicking normal cortisol secretion [5].

Recent studies suggest that the plasma concentration of hydrocortisone replacement therapy is heavily subjected to inter-individual variability, giving rise to side effects such as daily fatigue and an increased risk of cardiovascular disease [5]. In addition, studies have shown that the protein binding of hydrocortisone to transcortin (cortisol-binding globulin—CBG) and albumin can affect the pharmacokinetic parameters of hydrocortisone [6]. In order to develop and evaluate more appropriate substitution therapies, a better understanding of the pharmacokinetic properties of HC with protein binding is needed.

Approximately 80–90% of hydrocortisone binds to CBG, which can be found within the circulation [7]. Only the free fraction of hydrocortisone is responsible for its physiological function. Changes in the binding kinetics of hydrocortisone to CBG can potentially impact free hydrocortisone concentration, which is known to show inter-individual variability [8]. Calculating this could therefore be beneficial for treatment [9,10,11]. CBG can exist in an intact state (CBGi) or cleaved state (CBGe) [12]. CBGe has a 10-fold lower affinity to hydrocortisone than CBGi [13]. This state can be influenced by neutrophil elastase and chronic inflammation [14]. To understand and describe the pharmacokinetics of hydrocortisone, pharmacokinetic models can be used. An example of such a model is the model made by Werumeus-Buning et al. [5]. A drawback of the Werumeus-Buning model is that for total and for free hydrocortisone, separate models are developed. Combining both models could potentially increase their predictive power. This study focused on improving the pharmacokinetic model made by Werumeus-Buning et al. by including protein binding (albumin and CBGe-CBGi) [6] in order to optimize substitution strategies with hydrocortisone. This model was used to simulate a dosing strategy for patients suffering from SAI. This new dosing strategy is currently under development.

## 2. Materials and Methods

### 2.1. Subjects and Measurements

This study is a post hoc analysis of the data generated by Werumeus-Bunning et al. [5] by adding albumin and CBG binding to the model. The details of this study design have been published in detail previously; the patients and intervention are the same as described in the study by Werumeus-Buning et al. [5] and are summarized in brief here. The study was approved by the ethical committee of the University Medical Center Groningen, The Netherlands. All patients gave written informed consent before entering the study. The study was registered with ClinicalTrial.gov, NCT01546922.5.

Sixty-three patients with SAI were recruited for a randomized double-blind trial between May 2012 and June 2013 from the outpatient clinic at the University Medical Center Groningen. A total of 47 patients completed all study procedures. Due to a violation of the study protocol, one patient was excluded. All patients were on a stable glucocorticosteroid replacement therapy for at least six months prior to the study. Renal and liver parameters were tested and perceived to be normal in each subject. Patients that were known to use active CYP3A4 inhibitors or inducers were excluded. Further inclusion and exclusion criteria are published in detail elsewhere [5].

Patients were randomly allocated to receive a low dose of HC (0.2–0.3 mg HC/kg body weight/day) for ten weeks followed by a higher dose (0.4–0.6 HC/kg body weight/day) for ten weeks or in reverse order. HC (immediate release tablets) were given in three divided doses (before breakfast, before lunch, and before dinner). The exact dosing scheme has been previously described in detail [5]. Patients were allowed to double or triple their dose in case of illness. Patients were strictly instructed not to take a double or triple dose in the week preceding the visiting day. On visit days, patients were instructed to take their HC dose at 7:00 h and 8:00 h blood samples were drawn following a normal protocol (upright sitting, after a short period of rest). This procedure was repeated after approximately 5 h.

Plasma free hydrocortisone, total hydrocortisone and cortisone levels were measured using isotope dilution liquid chromatography tandem mass spectrometry (LC-MS/MS) at the laboratory of the UMCG. An internal standard of hydrocortisone 13C3 was used. The method that was used was described by Hawley et al. [15]. Plasma equilibrium dialysis for free cortisol and cortisone was performed as described by Fiers et al. [12], the only difference being that we used 10 kD cellulose membranes (Harvard Apparatus). Total serum CBG concentrations were measured in duplicate using a radioimmunoassay (IBL International GmbH, Hamburg, Germany) [16]. Albumin was determined with the bromocresol green method on a Roche Modular ISE/P (Roche Diagnostics, Mannheim, Germany).

### 2.2. Model Development

A 1-compartment pharmacokinetic (PK) model was developed using the object-oriented visual PK/PD modeling tool Edsim++ (version 1.88, Mediware, Prague, based on MwPharm++) [17,18,19] as is shown in Appendix A. With Edsim++ it was possible to develop a visual object-oriented model with an integrated Nguyen formula [20]. Previous analysis was also conducted using an earlier version of KinPop++ (Werumeus-Buning et al.), which is part of the Edsim++ PKPD modeling platform. Using KinPop++ would allow us to easily compare both models. A special compartment class (XCortisol) was developed for modeling the protein (albumin, CBGe-CBGi) binding of hydrocortisone. The visual Edsim++ representation of the model was exported to Berkely Madonna (Appendix A) and RxODE (Appendix A) for internal model validation purposes. This compartment model employs Equation 1 derived from Nguyen et al. [20], which is based on earlier work from Dorin et al. [21] for the estimation of bound (‘total’) and unbound (‘free’) hydrocortisone. Edsim++ was used for the population analysis of the free and total hydrocortisone plasma levels using an iterative two-stage Bayesian (ITSB) technique [22]. In Equation (1) K_A_ represents the affinity of albumin for hydrocortisone, *K_C_* represents the affinity for intact CBG (CBGi) for hydrocortisone and *K_C*_* represents the affinity of cleaved CBG (CBGe) for hydrocortisone. *TotF* = total concentration hydrocortisone. *F* = concentration hydrocortisone. *TotA* = total concentration of albumin. *TotC* = total concentration of CBGi. *TotC** = total concentration of CBGe.
(1)$$TotF\;=\;[F]\;\times \;(1\;+\;\frac{TotA}{K_{A}\;\times \;[F]}\;+\;\frac{TotC}{K_{C}\;\times \;[F]}\;+\;\frac{{TotC}^{*}}{K_{C^{*}}\;\times \;[F]})$$

The model can operate in two modes (free or total hydrocortisone concentration). In this study, we used the free hydrocortisone mode, in which hydrocortisone elimination is driven by the free concentration. The total concentration is then calculated from the free concentration.

The population values for the clearance (236 L/h) and volume of distribution (474 L) as found by Werumeus-Buning et al. [5] (Table 1) were used as first estimates in the new model. Bioavailability was set to a literature value of 0.96 [9,14]. Dissociation constants were derived from Nguyen et al. [20] and set as follows: *K_A_*, *K_C_* and *K_C*_* were set to 330,000 nM, 330 nM and 33 nM, respectively. The ratio between CBGe-CBG was individually predicted for every patient and is shown as an RC value, calculated as either CBGe = RC × CBG or CBGi = (1 − RC) × CBG.

By using Equation (1) and data from Werumeus-Buning et al. [5,6] it was possible to calculate new values for the clearance and volume of distribution that takes protein binding into account (Table 1). This model can therefore use both free and total hydrocortisone values in one model.

### 2.3. Iterative Two-Stage Bayesian

The ITSB method is currently the only method supported by KinPop and has been proven to perform well in both rich data sets [22] and in sparse data sets [23]. Moreover, an ITSB analysis and its individual estimates are hardly affected by most data registration errors [24].

### 2.4. Statistics and Evaluation

Statistics were carried out using GraphPad Prism v 8.0. Spearman’s r correlation was calculated to compare the observed versus predicted values. Significance was calculated with an unpaired *t*-test. To assess the validation of the obtained results, a goodness-of-fit plot and an accessory weighted residual graph were plotted.

### 2.5. Simulated Plasma Curves with a New Dosing Strategy

For the development of a new dosing strategy for SAI therapy with hydrocortisone, it was deemed most beneficial for patients that the hydrocortisone plasma concentration–time behavior be as similar as possible to the endogenous cortisol plasma values. To achieve a simulation of close-to-endogenous cortisol values, it was concluded that hydrocortisone should be dosed twice daily. One dose in the evening has a sufficient dissolution lag time to give a peak plasma concentration in the morning. The second dose is administered in the morning with sufficient slow absorption to sustain the hydrocortisone release during the day. From the pharmacokinetics, an evening dose of 9 mg was taken and a morning dose of 5 mg. For the evening dose, a dissolution lag time of 4 h was used and a fractional absorption rate of 0.7 h^−1^. For the slow absorption of the morning dose, a zero-order dissolution rate of 0.8 mg/h was chosen.

## 3. Results

### 3.1. Concentration of Hydrocortisone Bound to Protein

Figure 1 shows the binding state of hydrocortisone with CBG for a typical patient (weight; 70 kg, height; 1.75 m) receiving a dose of 10-5-2.5 mg at t = 7:00, 13:00 and 19:00 h (morning, midday, and evening) of immediate-release HC. Binding to CBG, expressed as the fraction unbound (fu), showed variation over time (dotted line). The concentration of hydrocortisone bound to protein (CBG and albumin) was simulated, as well as the free unbound form of hydrocortisone.

### 3.2. Comparing New Model to Older Model

The results in Table 1 show the estimated population pharmacokinetic parameters for total hydrocortisone in the new model. Our model included the estimated ratio of CBGe-CBG. Both the clearance and the volume of distribution were found to be slightly lower, although this result was not significant (*p* = 0.11). The difference in CL between both models was also not significant (*p* = 0.67). Interindividual variance was 55% for V, 51% for Cl and 85% for RC value. Additionally, the AUC_24_ values for total versus free cortisol were calculated and compared between the new model and the values of the older model.

### 3.3. Goodness-of-Fit Plots

In order to assess the performance of the model, goodness-of-fit plots and residual plots were generated for population and individual predicted values (Figure 2 and Figure 3). An overall correlation of r = 0.9985 could be found for the individual prediction with measured data, with a 95% confidence interval of 0.9981 to 0.9988. For population values, r = 0.9542 with a 95% confidence interval of 0.9430 to 0.9632. Residual plots are shown in Appendix A.

## 4. Discussion

### 4.1. Key Findings

A new population pharmacokinetic model of hydrocortisone was developed as an improvement upon previously developed models by considering the different affinities for hydrocortisone for the two forms of CBG:iCBG and eCBG. Because the protein binding of hydrocortisone plays a role in hydrocortisone kinetics, it is argued that this model gives a better prediction of free or total hydrocortisone in the human body.

### 4.2. Interpretation and Implications

Table 1 shows the model parameters for V and CL, including a mean RC value of 0.13, using the population model. This is in concordance with the literature [9] and is therefore a plausible outcome of the model. A lower volume of distribution was found for free hydrocortisone when comparing the new population calculations with the Werumeus-Buning [5,6] calculations. Because the free fraction is dependent on CBG binding, and CBG is mainly found in the bloodstream, a lower V is observed and expected in this new calculation. The new model has both free and total hydrocortisone in one model, whereas the old model had one model for free and one model for total hydrocortisone. With the new model, it is therefore possible to easily calculate unbound hydrocortisone values from total hydrocortisone values. When comparing AUC_24_ values, a higher correlation can be observed between free hydrocortisone AUC_24_ values versus total hydrocortisone AUC_24_ values. This implies that the model is in fact working as intended, as a higher correlation between the two values would be expected when protein binding is factored.

Figure 1 shows that the plasma values of hydrocortisone bound to CBG are higher than the unbound values, as is in concordance with literature [8,9,10,20]. The plot also shows changes in the unbound–bound fraction over time, and that this is not one static value.

Figure 2 shows the predicted values based on the population parameter values. A good model can predict individual values with high accuracy. For the population prediction the model has an r value of 0.9452. High overall correlation can be found (r = 0.9985) when comparing the observed values with the predicted values for the individual fit. Thus, the model is able to predict hydrocortisone plasma concentrations with high accuracy. This also gives credibility to the curve as is seen in Figure 1 for a typical patient, but also for every individually fitted patient. It is also possible to gain new information and calculate important parameters such as the area under the curve (AUC) with this model.

By including protein binding in our model, it permits a better evaluation of adrenal function (when measuring total cortisol and CBG), which is particularly useful in patients with altered CBG concentrations. Altered CBG concentrations can occur in patients who have high neutrophil elastase activity [10,12,19]. 

The simulated curve and conventional therapy curve in Figure 4 show that the conventional therapy lacks the early rise in cortisol between 3 and 6 h in the morning for this patient. It is therefore argued that an evening dose should have a sufficient lag time to mimic as closely as possible the endogenous cortisol plasma concentration.

### 4.3. Limitations

Some limitations of this study need to be addressed. First, total CBG (CBGi + CBGe) was measured, instead of CBGi alone. Therefore, the RC ratio needed to be estimated by the model rather than measured. Measuring CBG directly could lead to better results. However, as Table 1 shows, an estimated RC of 0.13 is plausible compared to the literature values of the RC ratio (as is addressed by Nguyen et al. [20]).

Furthermore, our model does not include factors such as steroid competition (as hydrocortisone is not the only steroid that can bind to CBG). Concentrations of steroids such as testosterone and progesterone may vary between individuals (as is described in Gudmand-Hoeyer et al. [25]), and in turn influence binding to CBG [14]. CBG also actively regulates hydrocortisone release [26]. Currently, this is not quantitatively described in the literature. Some study subjects also took steroid supplements, which could have influenced the result in this study.

CBG has a circadian rhythm. One could argue that the circadian rhythm of CBG can influence the hydrocortisone dosing strategy. However, Melin et al. [27] found that the difference in hydrocortisone exposure is ≤12.2% between the times of highest and lowest CBG concentrations; therefore, hydrocortisone dose adjustment based on the time of dosing to adjust for the CBG concentrations is unlikely to be of clinical benefit.

Inter-individual variability in the production of endogenous hydrocortisone could also affect the outcome of the model [28]. It is, however, still unclear if this is of clinical significance. The study conducted by Vulto et al. [29] described that adrenal corticosteroid production is likely to continue during treatment in a considerable percentage of patients with both primary and secondary adrenal insufficiency. However, these residual corticosteroid concentrations were quantitatively low and might not be of relevance.

CBG concentration can also be different when using oral contraceptives, as some studies found that CBG concentrations increased rapidly during the first cycle of treatment with oral contraceptives [30].

### 4.4. Recommendations

It is interesting to use this model to further investigate optimal hydrocortisone therapy in patients suffering from adrenal insufficiency. In normal subjects, endogenous hydrocortisone levels steadily rise from around 3:00–5:00 h and peak at around 7:00 [31]. With conventional hydrocortisone therapy, patients administer their daily dose early in the morning, usually at breakfast. As a result, patients administer their HC dose at an inappropriate time, which in turn decreases overall well-being, as was investigated by both Werumeus-Buning et al. [5] and Simon et al. [11].

Our new model can easily be used to simulate hydrocortisone serum concentrations based on drug formulations with different types of release profiles, e.g., for coated HC tablets with a dissolution lag time (modified release) as a substitution for conventional immediate-release HC therapy to further improve therapy in SAI patients. These data can then be used for in vitro studies to further optimize hydrocortisone formulations.

Using this model to investigate in vivo release characteristics can aid drug development. This development is currently in progress, as new therapeutic interventions are simulated with the help of this model.

## 5. Conclusions

A model is developed describing total hydrocortisone and free hydrocortisone by accounting for the protein binding of hydrocortisone to albumin, iCBG and eCBG. This model provides a more accurate fit than the previously used models for calculating the concentration of free or total hydrocortisone in the circulation.

This model can be used in future studies towards drug development, hydrocortisone pharmacokinetics and dosing simulations. Unbound cortisol can be calculated in a simple and reliable way by measuring total cortisol and CBG and permits a better evaluation of adrenal function, which is particularly useful in patients with altered CBG concentrations.

## Figures and Tables

**Figure 1 pharmaceutics-14-01161-f001:**
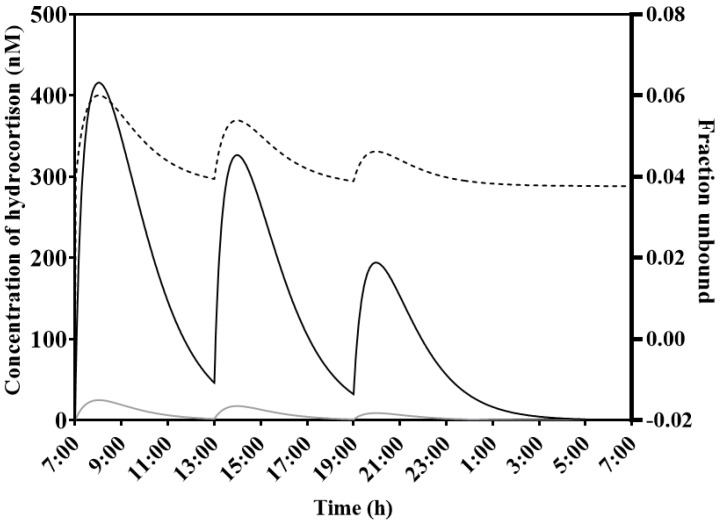
Simulation of free and total hydrocortisone plasma levels in a typical patient (weight; 70 kg, height; 1.75 m) receiving a dose of 10-5-2.5 mg of immediate-release HC at 7, 13 and 19 h. This figure shows the concentration of hydrocortisone that is bound to CBG and albumin (solid line, conc. bound—nM) and the concentration of hydrocortisone unbound (gray line) (conc. unbound—nM). Variable protein binding to CBG is also shown as a fraction (fraction unbound—dashed line, right axis).

**Figure 2 pharmaceutics-14-01161-f002:**
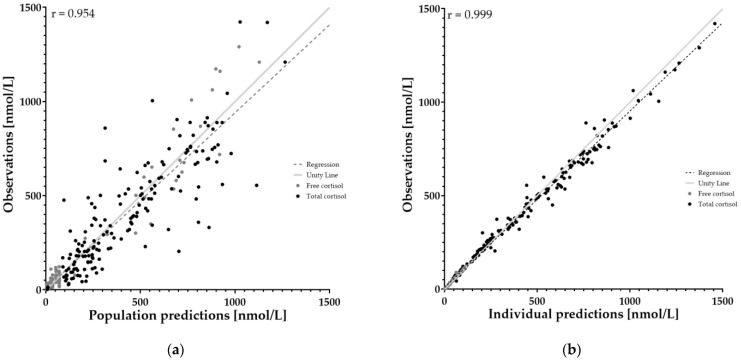
Goodness-of-fit plots of both population and individual parameters. This plot shows the agreement between observed and predicted concentration by model. Agreement is both visualized in the plots (red line as unity), as well as calculated using Spearman’s r correlation coefficient. Black color shows the total hydrocortisone concentration (bound to CBG and free). Grey shows the free hydrocortisone concentration. Spearman’s r for individual prediction is 0.999. Spearman’s r for population prediction is 0.954. (**a**) shows population prediction. (**b**) shows individual predictions.

**Figure 3 pharmaceutics-14-01161-f003:**
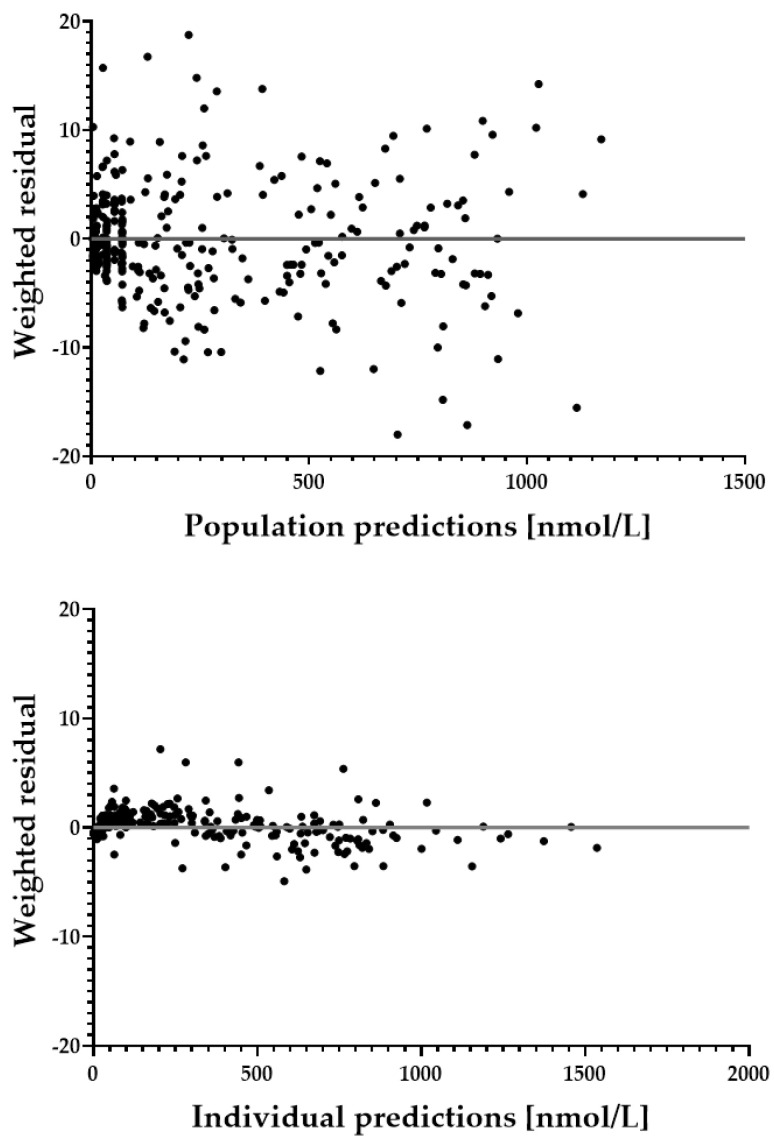
Weighted residual plots of both population (**upper**) and individual predictions (**lower**).

**Figure 4 pharmaceutics-14-01161-f004:**
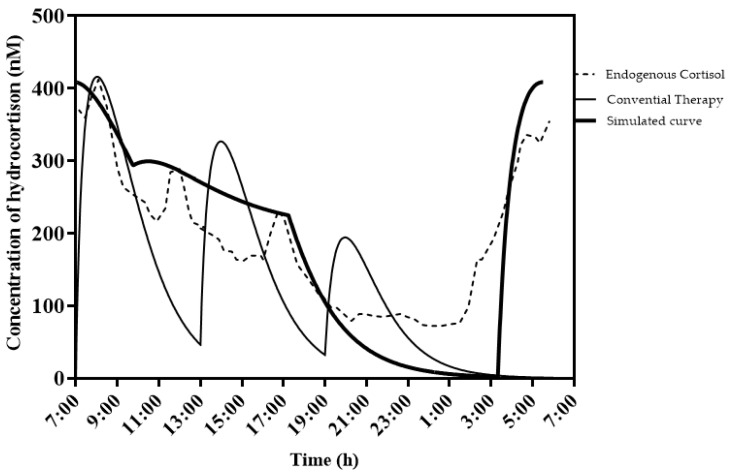
Simulated example of desired therapy (thick black line) with endogenous cortisol (dashed line) and conventional therapy (black line). Typical patient (weight; 70 kg, height; 1.75 m) receiving a dose of 10-5-2.5 mg of immediate-release HC at 7, 13 and 19 h. The simulated curve has two hydrocortisone doses, one in the evening with a lag time of approx. 4 h (23:00 h), and one in the morning that has an extendedrelease profile (at 7:00 h). Evening dose is 9 mg and morning dose is 5 mg. Details on the characteristics of both doses can be found in the text.

**Table 1 pharmaceutics-14-01161-t001:** Parameters of the population model developed from our patient data (high dose and low dose), and the free fraction of hydrocortisone. Residual variability is 1.1 mg/mL (additive) and 9.5% (proportional). Shrinkage is shown in %.

Parameter	New Model	CV (%) ^2^	Shrinkage (%)	Old Model [5,6]	CV (%)	*p*-Value (Old vs. New)
V (L)	405	55%	1.9	474	54%	0.11
CL (L/h)	226	51%	1.1	236	46%	0.67
RC ^1^	0.13	85%	19.8	-	-	-
Spearman’s r (AUC_24_ for free versus total hydrocortisone)	0.901	-		0.836	-	<0.0000001
Confidence interval	0.852 to 0.935	-		0.757 to 0.891	-	
*p* value (two-tailed)	<0.0001	-		<0.0001	-	
N	88	-		88	-	

^1^ RC = ratio between CBGe (corticosteroid-binding globulin cleaved) and CBG. ^2^ CV (%) is the interindividual variation present in both our model and that of Werumeus-Buning.

## Data Availability

Not applicable.

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
