# Peer review of "Pharmacokinetic Modeling of Hydrocortisone by Including Protein Binding to Corticosteroid-Binding Globulin"

_pharmaceutics, 2022, doi:10.3390/pharmaceutics14061161_

Round 1

Reviewer 1 Report

Rozenveld and co-authors have presented a modified PK model to predict HC the article is well-presented and provided sufficient data/conclusion to support the modified model. Although loads of exclusion criteria made in this study would be interesting to know the further effect such as age, concomitant therapy, have the authors investigated that already?

Author Response

Dear Reviewer,

Thanks for the comments on the paper, I appreciate the feedback and will implement these. The use of age and concomitant therapy is interesting and we were indeed planning on further studying this in upcoming studies. 

Kind regards,

Eric

Reviewer 2 Report

I appreciate my consideration as a reviewer regarding the manuscript “Pharmacokinetic Modeling of Hydrocortisone by Including Protein Binding to Corticosteroid-binding Globulin” by Rozenveld et al.  

Major comments:

  • I have basically only one major comment. This article is motivated by the need for suitable models to improve dosing with hydrocortison”. The model was successfully developed, but the recommendation is to use this model for simulations. Why did you not do simulations yourself? I would strongly recommend to do simulations and give therapeutic suggestions to give the article far more clinical impact.

Minor comments:

  • Keywords: Why PK/PD modelling? What is the PD aspect? Please explain or change to “pharmacokinetic modelling”
  • Row 44: “healthy life” is too general. Please adapt
  • Row 45-47: these two sentences are saying more or less the same. Please connect them to one sentence.
  • What is the disadvantage to have two models? Or: What is the advantage of having a combined model? Please add a short sentence/paragraph in the introduction
  • Methods: Edsim++ is not the Goldstandard of PopPK-Modelling. Why did you use this software? As the program is for most users (including me) rather unknown, please explain briefly how it works and why you used it.
  • 1: Abbreviations TotF and F are not explained. Please add this information.
  • Why did you use ITSB-technique?
  • Lines 146 to 148: Should probably be a komma-separated sentence? Abbreviations are not explained again and there are two “=” in one formula.
  • Line 175: What is RC?
  • Line 176: What was the result of the calculated AUCs?
  • Please add the Nguyens formula to the manuscript.
  • Methods: Some internal model diagnostics is missing here (VPX, NPDE). Please Add! GOF plots are not enough to assess quality of the model.
  • Figure 1, Caption: “Typical simulation” doesn’t make sense, please change to: “Simulation … of a typical patient”
  • Figure Caption 2: Why are there brackets on top of the Caption? In addition, line 213 should be: “between observed and predicted concentration”.
  • Table1: Residual variability/error is missing? Shrinkage is missing!
  • Line 236: “Key” (k in capital letter)!
  • Line 244-247: Should be moved to “results”
  • Line 269-271: Please correct. Sentence does not make sense as it is.
  • Lines: 319-321: This statement is not supported by your data, but clinically indeed very interesting. Please include in your article (Introduction, methods and results section) some clinical information on target/target attainment of your population and results of the simulations/individual predictions.

Author Response

Dear Reviewer.

Thanks for reviewing our manuscript. We appreciate the detailed feedback and you raised some strong points to improve the manuscript. See attachment for detailed description of what we did with the comments.

Kind regards,

Eric Rozenveld, Msc.

Reviewer 3 Report

The submitted manuscript deals with the elaboration and validation of a POP/PK model that could be able to analyze both free and bound hydrocortisone at the same time. Final findings suggested that the model adequately described and fitted measured concentrations of bound and free plasma hydrocortisone. Overall, the manuscript is interesting and represents a step forward with respect the previous models.

I have some minor queries for the authors.

First of all, I would see the residuals-vs-time plot, in order to judge the PK part of the model.

Second, a reference for values of dissociation constant (lines 145-146) should be added, as it has been done for values of other parameters.

Furthermore, I agree with the authors about the limitations of the study, especially when considering time-varying endogenous molecules (i.e., hydrocortisone, sex hormones, CBG). Although, the inclusion of such variables within the model could not bring new information, do the authors think that considering a circadian rhythm could improve the personalization of treatments with hydrocortisone? See for example the article published by Melin and colleagues (PMID: 32052005)

Please check consistency of acronyms throughout the manuscript.

Figure 1, legend. Please substitute length with height

Author Response

Dear Reviewer,

Thank you for reviewing our manuscript. These are some good points; the circadian rythm is very interesting for us, as we see in studies that are conducted with this model that this is indeed lacking in current therapy; for example, the rise in endogenous cortisol early in the morning. We ae currently working on a formulation which offers a 'lag-time'. this will hopefully be published in a second paper. 

I added the residual plots in the main text and the missing references. 

I substituted heigth with length for figure 1. 

Kind regards,

Eric Rozenveld Msc. 

Round 2

Reviewer 2 Report

Many thanks for replying to my comments.

I would like to point out one aspect (again):

1) regardless of a new formulation in developement, dosing simulations would be beneficial for this manuscript. Even if a new formulation is in developement, it will take years to a clinical utilisation of this new formulation and readers would be interested in dosing simulations using available formulations to optimise their day by day work. Therefore i recommend again to integrate dosing simulations into the present manuscript

I recommend again to consider whether it would be worthwhile to include this aspect in the manuscript to improve your work.

Author Response

Dear Reviewer,

Thank you for your message. We fully agree and are therefore excited to announce that we incorporated dosing strategy in the manuscript. It was deemed beneficial for the patients to include a sustained release tablet (Plenadren 5mg for example). A new formulation with sufficient lag time for early rise cortisol is currently in development. I attached the new manuscript.

Kind regards,

Eric
